# Design Optimisation of a Flat-Panel, Limited-Angle TOF-PET Scanner: A Simulation Study

**DOI:** 10.3390/diagnostics14171976

**Published:** 2024-09-06

**Authors:** Matic Orehar, Rok Dolenec, Georges El Fakhri, Samo Korpar, Peter Križan, Gašper Razdevšek, Thibault Marin, Dejan Žontar, Rok Pestotnik

**Affiliations:** 1Faculty of Mathematics and Physics, University of Ljubljana, 1000 Ljubljana, Slovenia; rok.dolenec@ijs.si (R.D.); peter.krizan@ijs.si (P.K.); 2Jožef Stefan Institute, 1000 Ljubljana, Slovenia; samo.korpar@ijs.si (S.K.); gasper.razdevsek@ijs.si (G.R.); dejan.zontar@ijs.si (D.Ž.); rok.pestotnik@ijs.si (R.P.); 3PET Center at Yale School of Medicine, New Haven, CT 06510, USA; georges.elfakhri@yale.edu (G.E.F.); thibault.marin@yale.edu (T.M.); 4Faculty of Chemistry and Chemical Engineering, University of Maribor, 2000 Maribor, Slovenia

**Keywords:** time-of-flight positron emission tomography (TOF-PET), GATE, Monte Carlo simulation, CASToR, PET detector development

## Abstract

In time-of-flight positron emission tomography (TOF-PET), a coincidence time resolution (CTR) below 100 ps reduces the angular coverage requirements and, thus, the geometric constraints of the scanner design. Among other possibilities, this opens the possibility of using flat-panel PET detectors. Such a design would be more cost-accessible and compact and allow for a higher degree of modularity than a conventional ring scanner. However, achieving adequate CTR is a considerable challenge and requires improvements at every level of detection. Based on recent results in the ongoing development of optimised TOF-PET photodetectors and electronics, we expect that within a few years, a CTR of about 75 ps will be be achievable at the system level. In this work, flat-panel scanners with four panels and various design parameters were simulated, assessed and compared to a reference scanner based on the Siemens Biograph Vision using NEMA NU 2-2018 metrics. Point sources were also simulated, and a method for evaluating spatial resolution that is more appropriate for flat-panel geometry is presented. We also studied the effects of crystal readout strategies, comparing single-crystal and module readout levels. The results demonstrate that with a CTR below 100 ps, a flat-panel scanner can achieve image quality comparable to that of a reference clinical scanner, with considerable savings in scintillator material.

## 1. Introduction

In positron emission tomography (PET), time-of-flight (TOF) information can be used to localise an event on the line of response according to the time difference in the detection of the two annihilation photons. The localisation accuracy is determined by the coincidence time resolution (CTR) of the detectors; for example, a modern Siemens Biograph Vision clinical scanner [1] achieves a CTR of 214 ps, which results in spatial localisation of the events of 3.2 cm. In laboratory setups, CTR values of 58 ps [2] and 32 ps [3] have been achieved. Scaling such technologies to clinical scanners may not be practically possible at the moment; however, the results indicate the potential to improve upon the currently used scanners. It is expected that the next generation of optimised TOF-PET silicon photomultipliers and electronics, which is currently under development, will enable 75 ps CTR values at the system level [4].

With sufficiently improved CTR, full angular coverage is no longer required [5,6]; thus, new detector geometries such as flat-panel detectors are possible, as confirmed by our previous studies [7]. One of the main benefits of flat-panel detectors over conventional ring detectors is the reduced cost. For example, the main studied flat-panel geometry uses 29% of the volume of scintillator material and 66% of the number of photodetectors and electronics channels of the reference cylindrical scanner. This geometry improves the possibility of modularity, for example, using a configuration with two or four panels or even extending the axial field of view (FOV) of the scanner by placing the panels consecutively. An example of such a geometry is shown in Figure 1. Scanners with flat-panel detectors would be smaller, lighter, more compact and mobile. Such new designs are explored in PetVision, a Pathfinder Open project of the European Innovation Council [8]. Other studies investigating flat-panel detectors have been performed, including total-body, two-panel scanners featuring monolithic detector designs with depth-of-interaction (DOI) capabilities [9,10]. Our study, on the other hand, focuses CTR improvements and cost reductions by using smaller total crystal volumes.

Improving the CTR requires optimisation at every level of detection, as each detector component contributes to the time resolution [11]. The material properties of the scintillator, such as its rise and decay time and its light yield, contribute 12 ps to the CTR in the best currently available scintillators (LSO:Ce:Ca), with a rise time of 20 ps and a decay time of 30 ns. LYSO:Ce crystals were measured with a rise time of 70 ps and a decay time of 40 ns, contributing 25 ps to the CTR [2]. The dimensions of the scintillator also contribute to the timing spread of the scintillation photons. The photodetector [12] and the electronics [13] both impact the time resolution of the system as well.

In this work, performed in the context of the PetVision project, simulation results for several designs of detectors used in a four-panel scanner are shown, and comparisons are made with the performance of a reference clinical scanner. Image quality was assessed using simulations of the NEMA phantom and according to the NEMA standard [14].

The aim of this study was primarily to explore the effects of various design choices for a flat-panel scanner configuration. The effects of the scintillator cross-section and length on the resulting image quality were examined. The effects of different CTRs of the entire system on image quality were also studied. The importance of crystal dimensions is trifold, since they affect the cost of such a scanner, the CTR and the DOI effect. As achieving state-of-the-art CTR is the objective, the chosen crystal length is below what would be considered standard, with the CTR compensating, to some degree, for the reduced sensitivity [7]. Besides the scanner parameters, different reconstruction parameters were studied. With the novel scanner configurations, investigation of different crystal readout strategies is of special interest. Basing the readout on the single-crystal level has the potential to provide the best possible spatial resolution; however, readout at multiple-crystal (module or submodule) levels has the potential for higher sensitivity by also capturing (some of) the inter-crystal scatter events. Some flat-panel scanner configurations, including the one studied in this work, enable higher spatial resolution by reducing the distance between the detectors and/or using a crystal with smaller cross-section. On the other hand, as discussed above, shorter crystal lengths are desired, so any gains in sensitivity need to be considered. Results obtained with two readout strategies at the crystal and submodule levels are compared in this work.

## 2. Materials and Methods

Simulations were performed using GATE software v8.2 [15], and the resulting images were reconstructed using CASToR v3.1.1 [16]. Due to the computationally intensive nature of both simulations and reconstructions, the Slovenian national supercomputing network (SLING) was used, mainly the HPC Vega [17]. Optical photons were not simulated; therefore, specifics of optical photon production, propagation and detection were not included in the simulation. These effects were only implicitly included as the final time resolution of the system, which was one of the inputs to the simulation. The simulated events were back-to-back gammas and not the decays of F18 or other $\beta^{+}$ emitters. During all reconstructions, only true coincidences were considered so that corrections for random and scattered events were not needed.

### 2.1. Scanner Parameters

The simulated scanner, shown in Figure 1, comprised four flat panels with dimensions of 30 × 30 cm^2^ and, in most cases, a fixed crystal pitch of 3 mm, meaning each panel contained 100 × 100 LYSO crystals (40,000 in total). For a specific study, an additional detector geometry with 150 × 150 detector elements and 2 mm pitch was evaluated. The distance between panels was set to 40 cm in all simulations. The geometric parameters were chosen as a compromise between cost and coverage of imaged phantoms. Our previous study showed the potential for good performance using such scanners [7]. The cross-section and length of the crystals were varied, as was the CTR of the system. The considered scintillator cross-sections were 3 × 3 mm^2^, 2.5 × 2.5 mm^2^, 2 × 2 mm^2^ and 1.5 × 1.5 mm^2^. The latter three resulted in a sparse configuration with empty space between the crystals, which leads to reduced geometric coverage and system sensitivity. Still, sparse configurations are of interest in this study, both in terms of reducing the cost of the scanner and potentially improving the CTR of the scanner due to improved light collection efficiency from the scintillator on the (larger) photodetector. It has also been demonstrated that sparse configurations are feasible without degrading clinical image quality, despite lower sensitivity when compared to a scanner with no gaps and more detector elements [18]. Simulations were performed for crystal lengths of 10 mm, 7.5 mm and 5 mm and CTRs of 75 ps, 100 ps, 150 ps and 200 ps. The effects of different reconstruction parameters were also studied.

To facilitate the presentation of results, the following naming scheme combining the characteristics of the scanners was employed: CS<cross-section>_L<length>_CTR<CTR>. For example, CS3_L10_CTR75 represents a configuration with a 3 × 3 mm^2^ cross-section, 10 mm length and 75 ps CTR. Crystal pitch was omitted in the names for all cases, except for the configuration with the 1.5 mm^2^ cross-section and 2 mm pitch. The readout level was also omitted from the naming scheme for crystal-level readout, while it was indicated for the submodule-level readout.

For the flat-panel scanner, the energy resolution was set to 10% at 511 keV, and the energy window was set between 435 and 585 keV. The coincidence window was set to 2 ns. Coincidences were formed at the crystal level, and coincidences from any pair of panels were allowed. The decision to form coincidences at the crystal level was made in order to fully study the characteristics of spatial resolution, even at the cost of scanner sensitivity.

Additional simulations were also conducted, in which coincidences were formed at the submodule level for a closer comparison with the reference scanner, with submodules as groups of 5 × 5 crystals. The winner-take-all energy readout policy was used.

For the reference scanner, the properties of the Siemens Biograph Vision (Siemens Healthineers AG, Forchheim, Germany) were used according to specifications reported in [1]. In short, the size of LSO crystals was set to 3.2 × 3.2 × 20 mm^3^, and crystals were grouped into submodules of 5 × 5. The submodules were then grouped in an array of 4 × 2 in modules, combined in arrays of 2 × 8 into sectors repeated 19 times in a ring with a lead shield at the edge. The bore diameter was 78 cm, and the axial FOV was 26.3 cm. In total, the scanner consisted of 60,800 crystals, with a total volume of 12.45 dm^3^. The energy resolution and window were set to the same values as for the flat-panel scanner. The CTR was set to 214 ps, and the coincidence window was set to 4.7 ns. Coincidences were formed at the submodule level with the take energy centroid policy. The minimum sector difference was set to 4.

### 2.2. Spatial Resolution

When using a flat-panel PET scanner configuration, the geometry is no longer cylindrically symmetric. Thus, the source position, as dictated by the NEMA standard, no longer sufficiently describes the spatial resolution across the FOV. In the new symmetry, it makes more sense to talk about resolution in the *X* and *Y* directions instead of radial and tangential directions so that they coincide with the voxels of the reconstructed image.

For this study, 125 spherical sources (referred to as point sources) with radii of 0.1 mm were simulated and arranged in a grid in the first octant of the FOV, forming a 5 × 5 × 5 cubic grid with 37 mm spacing between source positions, as shown in Figure 1. The distance was chosen in order for the sources to occupy most of the FOV in the first octant. Only one octant was used due to the symmetry of the scanner. The non-colinearity (GATE parameter accolinearity) of the source was set to 0.5°. Source activity was set to a low value of 70 Bq (0.45 kBq/mL) to avoid random coincidences. Each source was reconstructed using the maximum-likelihood expectation maximization (MLEM) algorithm with 30 iterations and using the distance-driven projector. Spatial resolution was evaluated using an iterative algorithm instead of the filtered back-projection algorithm due to the incomplete angular coverage of the flat-panel scanner. The voxel size was set to 0.8 × 0.8 × 0.8 mm^3^.

The reconstructed images were fitted with a Gaussian function expressed by the following equation:(1)$$f\left(x,y,z\right)=A\cdot e^{-\left(\frac{{\left(z-z_{0}\right)}^{2}}{2\sigma_{z}^{2}}+\frac{{\left(y-y_{0}\right)}^{2}}{2\sigma_{y}^{2}}+\frac{{\left(x-x_{0}\right)}^{2}}{2\sigma_{x}^{2}}\right)}+C$$

The resolution was reported as FWHM and calculated based on the standard deviations in the direction of each coordinate axis. One of the advantages of using this method to estimate spatial resolution over the NEMA standard is that the standard only considers a single line profile in determining the resolution in each of the directions. In contrast, fitting takes the entire reconstructed source into account. The NEMA method is sufficient when the source is reconstructed without any artefacts but may not be reliable in other cases. The disadvantage of the fitting method is that a reconstructed source may not have an exact Gaussian shape, so the fitted function may not perfectly represent the reconstructed source.

The influence of the number of iterations used in the reconstruction was also examined. Due to edge artefacts, as the number of iterations increased, simulations were repeated with point sources against a background of lower activity, which reduced artefacts due to the non-negativity constraint of the iterative reconstruction algorithm [19]. The background was represented as a homogeneous cube with a 4 cm side and 28.8 kBq (16.7 MBq/mL) of activity. The value was chosen to achieve a signal-to-background ratio of about 8 in the reconstructed images, making the signal easy to identify without making the simulations even more computationally intensive. The computational costs of both simulations and reconstructions are further addressed in the discussion Section 4. The acquisition time was varied from 30 to 133 h to compensate for the sensitivity of different scanner configurations.

Separate simulations were performed in which only a single source in the background was simulated at a position 10 mm from the centre in order to avoid geometric effects on the spatial resolution, as specified by the NEMA standard. Reconstruction was performed using different voxel sizes and numbers of iterations to determine the effect of each. A total of 100 iterations were performed, and the voxel size was set to 0.4 × 0.4 × 0.4 mm^3^, 0.6 × 0.6 × 0.6 mm^3^ and 0.8 × 0.8 × 0.8 mm^3^.

### 2.3. Image Quality

Assessment of image quality for different scanner parameters was performed according to the NEMA standard [14], with the activity concentration set to 5.3 kBq/mL for the background and 21.25 kBq/mL for the source, making the ratio between them 4. The simulations were performed with the NEMA phantom, and 1 min acquisitions were collected for each configuration. Images were reconstructed with the MLEM algorithm using 50 iterations. The voxel size of the reconstructed images was set to 3 × 3 × 3 mm^3^, and images were upsampled by repeating each voxel 10 times in the *X* and *Y* directions, effectively splitting each voxel into a 10 × 10 grid of voxels with the same value. This was done to correct for the partial volume effect during analysis. Values of the percent contrast and background variability were calculated for the images with different Gaussian filters applied to them. The procedure was repeated 10 times in order to determine the statistical error. The same procedure was also repeated with the reference scanner.

Additional correction to normalisation were performed by simulating a homogeneous cube of activity in air. The dimensions of this cube were 36 × 36 × 36 cm^3^, filling the entire reconstructed FOV of the scanner. To avoid introducing additional noise to the reconstructed image of the NEMA phantom, a wide 15 mm Gaussian filter was applied to the homogeneous image. The use of a wider filter is justified by the gradual variation in sensitivity across the image. The image of the NEMA phantom was then divided by the homogeneous image and multiplied by its mean value.

### 2.4. Sensitivity

The sensitivity of the scanners was studied by simulating a point source at the centre of the FOV and moving it along the axial direction from one end of the scanner to the other in 1 mm increments. The number of counts was recorded and divided by the simulation time and activity of the source.

### 2.5. Noise-Equivalent Count Rate

The noise-equivalent count rate (NECR) metric was obtained by simulating a cylindrical polyethylene phantom containing a line source with a radius of 1.6 mm. The radius of the phantom was 10 cm, and the length was 70 cm. NECR was calculated using the following equation [14]:(2)$$\mathrm{NECR}=\frac{T^{2}}{T+S+R}$$
in which *T*, *S* and *R* are count rates of true, scattered and random coincidences, respectively, known exactly from the simulation. The simulation was performed at different activity values. The impact of crystal dimensions on the metric was evaluated. CTR also has an effect on the gain in the signal to noise ratio [20], which is characterised in effective NECR (${\mathrm{NECR}}_{\mathrm{TOF}}$) by the following equation:(3)$${\mathrm{NECR}}_{\mathrm{TOF}}=\frac{2D}{c\Delta t}\mathrm{NECR}$$
where *D* is the diameter of the imaged object, Δt the time resolution and *c* is the speed of light.

### 2.6. Brain Phantom

To facilitate qualitative assessment of the flat-panel system, which is particularly relevant for the end user, simulations were performed with a high-resolution digital brain phantom based on the BigBrain atlas [21]. The spatial resolution of this phantom is 0.4 mm, and it consists of high-resolution structures, making it a good choice for simulations of systems where improvements in spatial resolution can be observed. The simulated imaging time was kept constant for all scanners. The images were reconstructed with the MLEM algorithm using 50 iterations. The voxel size was set to 1 × 1 × 1 mm^3^. The structural similarity index [22] was used to compare a slice of the reconstructed image with the activity phantom in order to compare the performance of the CS3_L10_CTR75 scanner geometry with that of the reference scanner. The images were first cropped only to contain the relevant portions (a tight rectangle around the skull). A mean of two slices was considered for the activity phantom, since the slice thickness was lower.

## 3. Results

### 3.1. Spatial Resolution

#### 3.1.1. Effect of Number of Iterations on Spatial Resolution

To study the effect of the number of iterations, a point source with a background was simulated 10 mm from the centre of the FOV in the *Y* direction, placing it according to the NEMA standard. The resulting spatial resolutions in all three directions for voxel sizes of 0.8 × 0.8 × 0.8 mm^3^, 0.6 × 0.6 × 0.6 mm^3^ and 0.4 × 0.4 × 0.4 mm^3^ are shown in Figure 2. In all cases, up to 100 iterations were used in the reconstruction.

At a glance, there is good agreement between a higher number of iterations and improved spatial resolution, but it is important to note that in iterative reconstruction algorithms, especially when dealing with point sources, a higher number of iterations results in edge artefacts. A total of 30 iterations were run for the reconstructions in the rest of the spatial resolution study, since at that point, no edge artefacts can be seen.

#### 3.1.2. Spatial Resolution across the FOV

The results for the spatial resolution study are expressed by three parameters ($\sigma_{x}$, $\sigma_{y}$ and $\sigma_{z}$) for each of the 125 sources, making visualisation of all of the results difficult. To make the presentation easier, results are shown in two ways. First, the spatial resolution in all three directions is shown for all sources in a single slice of the FOV at a fixed axial coordinate. Results are always only shown for the central slice, meaning that the z coordinate is 0 (axial centre). The dependence of resolution on the source position in the XY plane at a fixed axial position of Z=0 for the CS3_L10_CTR75 configuration is shown in the top part of Figure 3, with the left plot showing the resolution in the *X* direction, the central plot showing the resolution in the *Y* direction and the right plot showing the resolution in the *Z* direction. A dark line indicating the approximate dimensions of the human head based on the phantom data used in this study is drawn to indicate the most relevant part of the FOV.

In the bottom part of Figure 3, the dependence of the spatial resolution on the axial position of the slice (Z=0 mm to Z=148 mm) for the CS3_L10_CTR75 configuration is shown for three source positions in the slice (at the centre of the slice and 111 mm from the centre in the *X* and *Y* directions); again, the left plot shows the resolution in the *X* direction, the central plot shows the resolution in the *Y* direction and the right plot shows the resolution in the *Z* direction.

The 2D histograms show that, as expected, the resolution deteriorates as the source is moved away from the centre of FOV towards a panel. The exception is the source placed at the very centre, where the resolution is also degraded by geometric effects (as the source is placed in the centre between the voxels of the reconstruction grid). Figure 3 also confirms that there is, in fact, symmetry in the *X* and *Y* directions, which would be observed if the resolution were presented as radial and tangential.

In the *X* and *Y* directions, the measured resolution ranges from 1.9 mm to 5.1 mm. Axial resolution ranges from 1.6 mm to 2.0 mm, and the worst result is confined to the source in the exact centre of the FOV. The bottom of Figure 3 shows that results remain fairly consistent at different axial positions and are all within the range of approximately 0.3 mm.

The deterioration of resolution in the *X* and *Y* directions at the edge of the FOV is mainly caused by the fact that very few lines of response connect two of the panels and pass through sources at the edge of the FOV. Even with traditional cylindrical scanners, a similar deterioration of spatial resolution is observed, and the FOV is restricted to a smaller area within those scanners (although in that case, this is due to the parallax error).

Figure 4 presents 2D histograms of the spatial resolution in the *X* direction in central slices for different scanner configurations. In sparse configurations, reducing the crystal cross-section improves the spatial resolution close to the centre of the FOV, especially if the pitch is reduced as well. Reducing the crystal length improves spatial resolution throughout the FOV, indicating the importance of DOI for flat-panel configurations. The best results were achieved with the CS3_L5_CTR75 system, where the resolution ranged from 1.6 mm to 2.7 mm. Comparing the results for the same scanner with different readout levels, the results show a deterioration of about 0.2 mm for the submodule-level readout compared to the crystal level.

For the reference scanner, the results range between 3.2 mm and 4.5 mm. Compared to the CS3_L10_CTR75 system, the mean resolution in the *X* direction is 0.8 mm worse for the reference scanner in the central slice. Unlike for the flat-panel geometry, the final layer of sources (placed 148 mm from the centre in the axial direction) falls outside the axial FOV of the reference scanner.

It is important to note that in all cases, the number of iterations and voxel size were fixed to make the comparisons fair, even though optimising the performance of each system might require the individual fine tuning of parameters for each case.

### 3.2. Image Quality

An example of a reconstructed image of the NEMA phantom can be seen in the top-left part of Figure 5. For visual comparison, the image reconstructed with submodule-level readout is also shown in the top centre of the figure. A lower level of noise can be qualitatively seen in the image from the submodule-level scanner when compared to the crystal level. To illustrate the importance of the CTR, an image reconstructed without TOF is shown in the top right of the figure. Reconstructions using CTR values of 100 ps, 150 ps and 200 ps are shown in the bottom of the figure to demonstrate the progression of the effect of improving the CTR.

For the sake of brevity, only the 28 mm and 13 mm spheres of the NEMA phantom are discussed. The results of contrast recovery versus background variability for various scanner configurations and for the reference scanner are shown in Figure 6. An experimental result from [1] for the reference scanner is also represented as a grey circle in all plots.

The left column of Figure 6 shows how a sparse design affects the image quality. This was achieved by keeping the pitch at 3 mm but changing the cross-section of the crystals. A configuration with a 2 mm pitch and 1.5 mm crystals is also considered. Decreasing the cross-section from 3 mm to 2.5 mm has little effect on contrast but deteriorates the variability. Decreasing the cross-section to 2 mm has a much more noticeable impact. The results show that at any given background variability, the submodule-level readout achieves a higher level of contrast than the crystal-level readout for the same scanner geometry and vice-versa.

The middle column of Figure 6 shows the effect of CTR. The improvements due to CTR are clearly visible.

The right column of Figure 6 shows the effect of crystal length. The image quality degrades with decreasing crystal length, which can mainly be attributed to a lower number of detected events due to reduced gamma stopping power, as the degradation is mainly apparent in background variability—in other words, due to noise.

### 3.3. Sensitivity

The influences of different lengths and cross-sections of scintillators on the system sensitivity are shown in the left part of Figure 7 compared to the reference scanner. It can be observed that the reference scanner outperforms all simulated configurations of the flat-panel scanner, but it is relevant to mention that for the flat-panel scanner, the coincidences are formed at the crystal level, while for the reference scanner, they are formed at the submodule level. To demonstrate the effect of this parameter, the CS3_L10_CTR75 scanner was simulated again, and coincidences were formed at the submodule level (labled with CS3_L10_CTR75submod), leading to more comparable results. The submodule-level readout of the flat-panel scanner improves the sensitivity by almost a factor of 2 compared to the crystal-level readout. Reducing the amount of scintillator material in sparse geometries and reducing crystal length both lead to reduced sensitivity. It should be noted that the CTR gain is not taken into account for this metric, and the smaller axial FOV of the reference scanner is also noticeable.

### 3.4. Noise-Equivalent Count Rate

Results for the dependence of ${\mathrm{NECR}}_{\mathrm{TOF}}$ on activity are shown in the right part of Figure 7, where different sparse configurations and crystal lengths are compared. The results for the reference scanner are included, and all values account for the gain due to TOF, as in Equation (Equation 3), where a value of 75 ps was chosen for the flat-panel scanner. Since this gain is a constant factor that depends only on CTR, there is no need to repeat simulations with different CTRs. Values of ${\mathrm{NECR}}_{\mathrm{TOF}}$ are the highest for the CS3_L10_CTR75submod, which takes advantage of both high sensitivity and a great CTR, but even the CS3_L10_CTR75 is able to outperform the reference scanner due to TOF gain, even though coincidences are formed at the crystal level and sensitivity is lower. In all cases, the amount of scintillator material is the most important factor, not the cross-section and length of the crystals.

### 3.5. Brain Phantom

A slice of the activity phantom in the transverse plane and the corresponding reconstructed images for the CS3_L10_CTR75 and the reference scanner are shown in Figure 8.

Despite the fact that the scanner is not specifically optimised for brain imaging—for example, the crystal cross-section is large for this application—brain images are well reconstructed, with many features visible. Similarly, the reference scanner is also not optimised for brain imaging and is also mostly not used as such in clinical practice, making the comparison somewhat unfair. However, the purpose of including it was to compare the performance of the scanners on an anatomical phantom with a high degree of detail.

The structural similarity index between the slice of the activity phantom and the corresponding image reconstructed by the CS3_L10_CTR75 scanner is 0.57. In comparison, with the reference scanner and the same acquisition time, the result is 0.51. The results obtained with varying Gaussian filters are shown in Figure 9, which indicates the convergence of the index for different systems as the filter width exceeds 5 mm. At lower filter widths, the crystal-level readout scanner performs better than the submodule level, as the spatial resolution is better, despite the worse sensitivity.

## 4. Discussion

The most important results presented in this article are summarised in Table 1, which shows the results for the spatial resolution, sensitivity, ${\mathrm{NECR}}_{\mathrm{TOF}}$ and percent contrast for different scanner parameters. The relative scintillator volume for each configuration is also included, comparing the performance to that of the reference scanner and one of the main indicators for the cost of the scanner. It is important to reiterate that the bore diameter of the reference scanner is 78 cm, while the panels are placed at a 40 cm distance. Moving the panels farther apart, for example, to 80 cm, is part of our ongoing research.

The study of spatial resolution showed that the cross-section of the crystal does not significantly impact the mean resolution in the central slice, but it does affect resolution in each individual location, most notably concerning the artefact due to symmetry at the centre of the FOV. Crystal length, on the other hand, was shown to have a major impact on the spatial resolution, with smaller crystals performing better. Measurement of depth of interaction is expected to improve spatial resolution without sacrificing sensitivity, which is the subject of our following work. CTR was also shown to have an effect on spatial resolution. In all cases, the mean spatial resolution for the flat-panel systems was better than for the reference scanner. On the other hand, the results for the reference scanner varied less across the central slice.

It is also important to mention the decision to simulate sphere sources with backgrounds. This study used iterative reconstruction algorithms in place of filtered back-projection for the assessment of spatial resolution due to the incomplete angular sampling of the flat-panel scanner. Iterative reconstruction leads to artefacts when reconstructing point sources, which become apparent after 30 iterations and may even lead to multiple peaks, degrading the resolution measure. The artefacts are exacerbated with decreasing voxel size. Adding the background activity cube was a way to reduce this issue, and artefacts start to appear at higher numbers of iterations. The benefits were discussed, but on the other hand, the method results in worse results for spatial resolution (which may, in fact, be more realistic); more importantly, it increases the complexity and computational cost by a large margin. For a single source without background, the simulation time is about 2 core hours, that for sensitivity image generation 24 core hours and that for reconstruction is 72 core minutes. For the same scanner with spherical sources with additional background activity, the simulation time is 140 core hours, there is no change for sensitivity image generation and reconstruction takes 23 core hours. In total, it is 27 versus 187 core hours per reconstruction. It is clear that without extensive use of grid computing, such a study would not be feasible. Other methods may be more practical, such as analysis of the edge response function [23,24], which is also not covered by the NEMA standard but allows for the reuse of reconstructed images from different studies, for example, the NEMA image quality phantom or a cylindrical phantom, instead of requiring separate point-source studies.

For the ${\mathrm{NECR}}_{\mathrm{TOF}}$ analysis, it was shown that in the best case, the flat-panel scanner performs better than the reference scanner. This is due to the greatly improved CTR at the system level assumed in this study.

In terms of image quality, it was shown that in the best-case scenario (CS3_L10_CTR75 scanner), the flat-panel scanner performs comparably to the reference scanner. It was also shown that a 2.5 mm wide crystal at a pitch of 3 mm does not substantially deteriorate the quality of the image, despite the 30% reduction in scintillator volume and the associated decrease in sensitivity. A further reduction of crystal cross-section to 2 mm leads to a much more severe deterioration.

If we compare the volume of the scintillator, which is one of the main contributions to the cost of a scanner, and take into account the flat-panel design that uses the most material, that is CS3_L10_CTR75, the flat-panel design uses only 28.9% of the scintillator material of the reference scanner. It is also important to note that despite this, the flat-panel scanner using this design has an axial FOV of 30 cm compared to the 26.3 cm of the reference scanner.

Regarding the comparison of crystal-level and submodule-level readouts, the results show that the decrease in spatial resolution is minimal (0.2 mm degradation from about 2.5 mm resolution), while the increase in sensitivity is significant (a factor of almost 2). This is reflected in the image quality metric, as both contrast and variability are lower; however, at any fixed value of either, it performs better than the crystal-level readout. The same is also reflected in the lower value of the structural similarity index for the brain images. Therefore, both modes have specific advantages. If a scanner was designed in such a way as to enable saving of the data obtained with both crystal readout strategies, images could be reconstructed for the highest spatial resolution as well, as the highest signal to noise ratio. Scanners with such capability may bring additional value to the modality in clinical practice.

## 5. Conclusions

In this work, we demonstrated that a flat-panel PET scanner can perform at the image quality level of current clinical scanners, given that a sufficient CTR is achieved at the system level. The improved CTR compensates for the reduced amount of scintillator material in the flat-panel scanner and prevents artefacts caused by the reduced angular coverage. The studied four-panel scanner with crystals of 10 mm length and a cross-section of 3 mm surpassed the reference clinical scanner in image quality metrics when simulated with a CTR of 75 ps and achieved better spatial resolution in most of the FOV, all with only about one-quarter of the total scintillator crystal volume.

Another interesting direction of research is a larger axial coverage. Flat-panel geometry makes this more easily achievable, as panels could be arranged in various configurations, including total body coverage, while only using as much scintillator material as the reference scanner. This will be the subject of our future research.

To better evaluate the spatial resolution in the flat-panel scanner geometry, computationally intensive simulations of a grid of point sources with and without uniform backgrounds were performed. The addition of a uniform background reduced artefacts in the reconstructed shapes of the sources and improved the reliability of the results. This was of special importance, since the flat-panel geometry enables reduced distance between detectors and, thus, better spatial resolution. For this reason, two crystal readout strategies—at the crystal and submodule levels—were investigated. The usual submodule-level readout achieves better sensitivity and image quality. The crystal-level readout provides limited benefits in terms of spatial resolution but could be used to provide additional information to the base, submodule-level readout image.

## Figures and Tables

**Figure 1 diagnostics-14-01976-f001:**
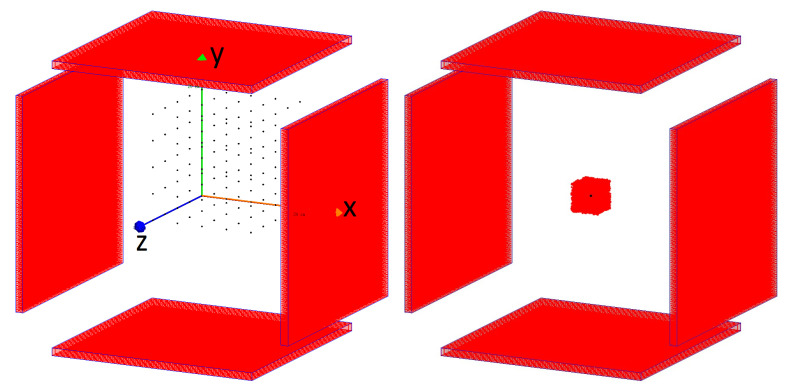
GATE visualisation of a simulated flat-panel scanner. The system consists of four panels with dimensions of 30 × 30 cm^2^ spaced apart by 40 cm. Each panel consists of 100 × 100 LSO crystals. The placement of spherical sources is also shown. On the left of the figure, sources are shown simultaneously, although they were simulated separately, and on the right, a single source in the centre of the FOV is shown with a background.

**Figure 2 diagnostics-14-01976-f002:**
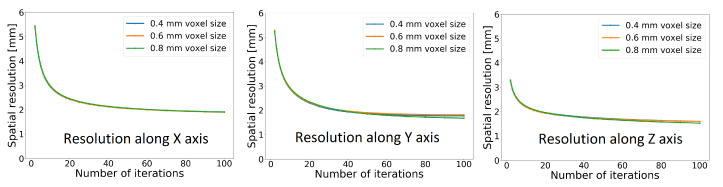
Dependence of the spatial resolution on the number of iterations at different voxel sizes for the CS3_L10_CTR75 scanner. The source position is 10 mm from the centre of the FOV in the *Y* direction.

**Figure 3 diagnostics-14-01976-f003:**
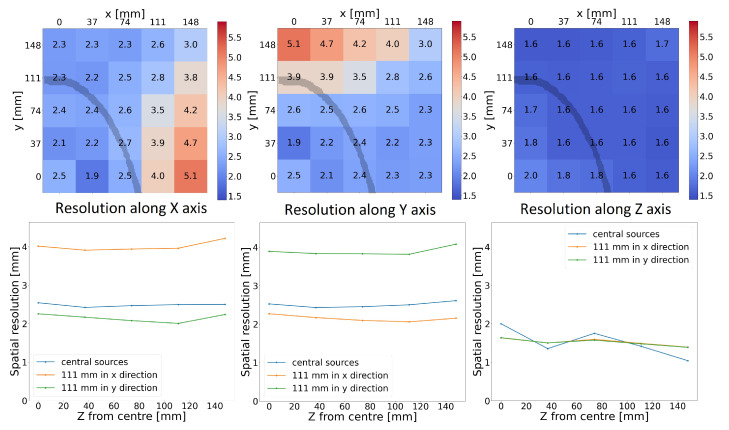
Spatial resolution for the CS3_L10_CTR75 scanner. Top: Spatial resolution on each axis for the same central slice. An outline of a skull is superimposed onto the image to show the relevant FOV for imaging. Bottom: dependence of spatial resolution on axial position for different *X* and *Y* positions.

**Figure 4 diagnostics-14-01976-f004:**
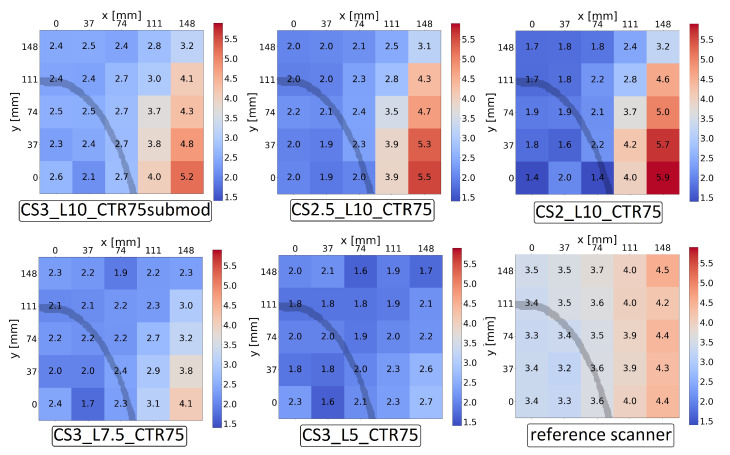
Spatial resolution in the *X* direction for a slice at the centre of the FOV for different geometries. An outline of a skull is superimposed onto the image to show the relevant FOV for imaging.

**Figure 5 diagnostics-14-01976-f005:**
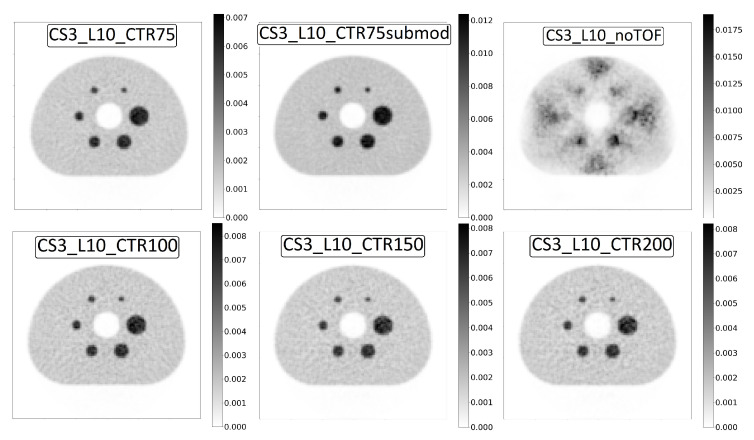
Reconstructed NEMA phantom. A 5 mm filter is applied to all of the images.

**Figure 6 diagnostics-14-01976-f006:**
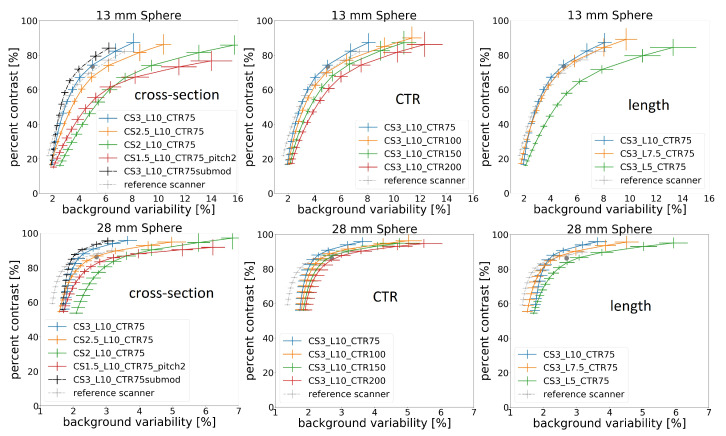
Image quality plots for the 13 mm and 28 mm spheres for different scanner parameters. The plots on the left compare the performance of different crystal cross-sections and pitches, the middle plots compare the CTR, and the right plots compare the crystal lengths. The grey circle represents the experimental measurement value for the reference scanner.

**Figure 7 diagnostics-14-01976-f007:**
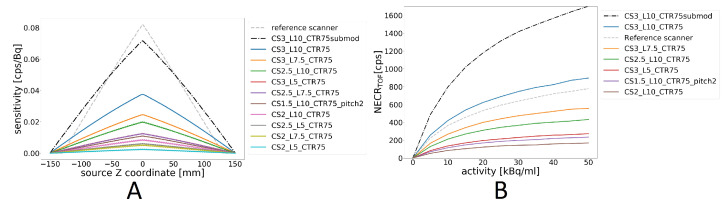
(**A**) Sensitivity profiles for different scanner configurations. ${\mathrm{NECR}}_{\mathrm{TOF}}$ plots comparing the crystal cross-sections and lengths (**B**) with the reference scanner. A configuration with readout at the submodule level (5 × 5 crystals) is also shown (CS3_L10_CTR75submod).

**Figure 8 diagnostics-14-01976-f008:**
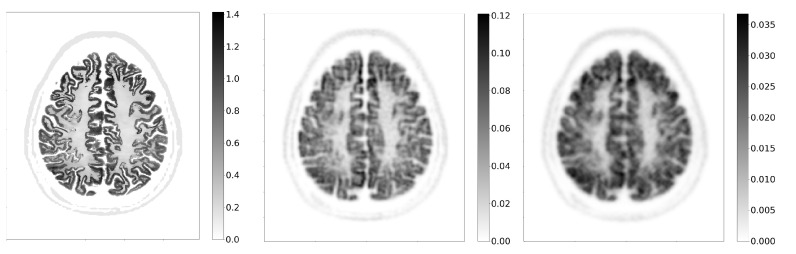
Images of the brain phantom in the transverse plane. On the left is the activity phantom, in the middle is the image reconstructed with the CS3_L10_CTR75 scanner and on the right is the image reconstructed with the reference scanner.

**Figure 9 diagnostics-14-01976-f009:**
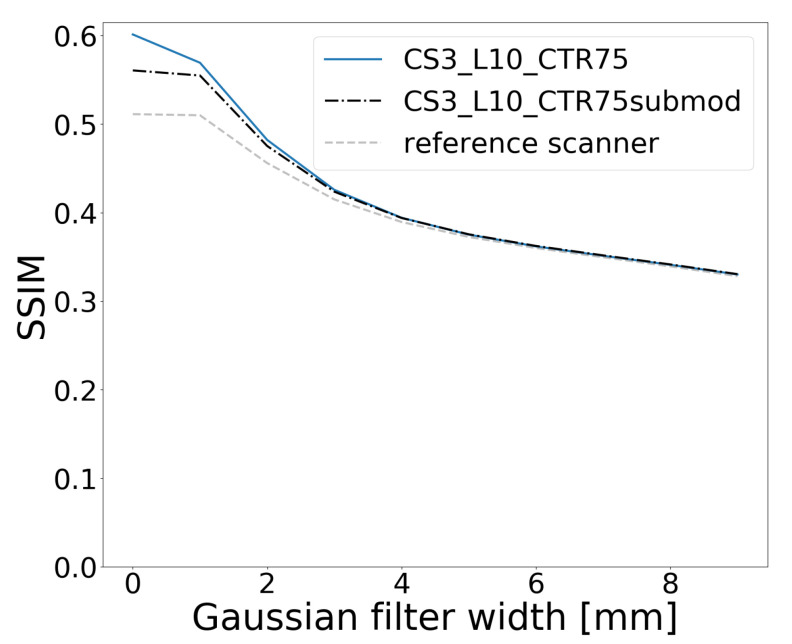
Dependence of the structural similarity index on the width of the Gaussian filter applied to the brain image.

**Table 1 diagnostics-14-01976-t001:** Summary of main performance metrics for different scanner parameters, including the relative scintillator volume compared to the reference scanner, spatial resolution in the *X* direction for a source that is located 37 mm away from the centre of the FOV in the *X* direction, total system sensitivity, ${\mathrm{NECR}}_{\mathrm{TOF}}$ at 10 kBq/mL activity and percent contrast at 5% background variability for a 13 mm sphere. The sensitivity and ${\mathrm{NECR}}_{\mathrm{TOF}}$ columns also contain relative values compared to the reference scanner in parentheses.

Geometry	Relative LSO Volume	Spatial Resolution [mm]	Sensitivity [kcps/MBq]	NECR_TOF_ [cps]	Percent Contrast
Reference scanner	100%	3.3	15.4 (100%)	365 (100%)	71%
CS3_L10_CTR75	28.9%	1.9	8.53 (55%)	420 (115%)	74%
CS3_L10_CTR75submod	28.9%	2.1	16.3 (106%)	798 (219%)	78%
CS2.5_L10_CTR75	20.1%	1.9	4.43 (29%)	212 (58%)	68%
CS2_L10_CTR75	12.8%	2.0	1.85 (12%)	85 (23%)	48%
CS1.5_L10_CTR75_pitch2	16.3%	1.3	2.39 (15%)	115 (32%)	53%
CS3_L10_CTR100	28.9%	2.0	8.53 (55%)	315 (86%)	70%
CS3_L10_CTR150	28.9%	2.2	8.53 (55%)	210 (57%)	65%
CS3_L7.5_CTR75	21.7%	1.7	5.55 (36%)	268 (73%)	72%
CS3_L5_CTR75	14.5%	1.6	2.81 (18%)	135 (37%)	56%

## Data Availability

The datasets presented in this article are not readily available because of technical limitations of data size. The size of simulation and reconstruction data used for this contribution is on the order of 10 TB.

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
