# Peer review of "Design Optimisation of a Flat-Panel, Limited-Angle TOF-PET Scanner: A Simulation Study"

_diagnostics, 2024, doi:10.3390/diagnostics14171976_

Round 1
Reviewer 1 Report
Comments and Suggestions for Authors
The manuscript discusses the performance of a flat panel limited angle TOF-PET system as a function of crystal size and length, gaps between crystals, crystal read-out strategy and time-of-flight temporal resolution. The discussion is based on GATE simulations and CASToR MLEM reconstructions. Results are compared to simulations of a Siemens Biograph Vision clinical PET/CT system.
The proposed flat panel PET/CT system was introduced in a paper published in IEEE Transactions on Radiation and Plasma Medical Sciences by the same research group in 2021. The present manuscript adds data and simulations to the previous work.
Comments:
P1: l21-23: It is ok to use the Biograph Vision Scanner as reference for the performance of a modern clinical PET/CT system, but the Vision scanner is not “the state-of-the-art” clinical system. Other systems, e.g. the United Imaging uMI Panorama PET/CT have better performance specifications, including CTR (189 ps) (cf. “Performance Evaluation of the uMI Panorama PET/CT System in Accordance with the National Electrical Manufacturers Association NU 2-2018 Standard. Guiyu Li, Wenhui Ma, Xiang Li, Weidong Yang, Zhiyong Quan, Taoqi Ma, Junling Wang, Yunya Wang, Fei Kang and Jing Wang. Journal of Nuclear Medicine April 2024, 65 (4) 652-658; DOI: https://doi.org/10.2967/jnumed.123.265929). I therefor suggest rephrasing of this sentence.
P2-3: subsection “2.1 Scanner parameters”: I miss some details in this section on how corrections for scatter, attenuation, randoms, and normalisation are implemented. In theory, data are directly available from the simulation. However, image quality depends on PET corrections and implementation details are important.
P3: l111-119: Please, specify explicitely both the axial and transaxial FOVs of the reference scanner. Obviously the volume and cost of scintillators in the reference system are higher than on the panel system , but that seems to me to be mainly because of a larger gantry opening – 78 cm vs. 40 cm.
P3: l131: Why is it important to avoid randoms in the simulation, if the contribution to the sinograms are completely known from simulation? Again, I miss more details on the corrections.
P3: l133: MLEM reconstructions are used for evaluating spatial resolution, which is not according to the NEMA NU-2 2018 specifications, which specify FBP. I guess FBP is not used because of gaps in the projection set between dectector panels. Nonetheless, please give some details on why FBP was not used.
P4: l147: Convergence and artefacts from the MLEM algorithm will depend on the number of iterations, signal-to-background contrast, and statistics. Please specify the acquisition time?
It is specified that the background activity was 28.8 kBq in a volume of 4*4*4 cm3. It is also specified that the source activity was 70 Bq in a voxel size of 0.8 mm (should be specified as 0.8x0.8x0.8 mm3). What was the reconstructed background voxel activity concentration (kBq/ml)? The source activity is 70 Bq. What was the reconstructed source activity voxel concentration (kBq/ml)? I cannot from these numbers alone reproduce a source-to-background ratio of 8. Maybe the specified source-to-background ratio is what you measure on your reconstructed images?
I suggest that you rephrase and clarify details.
P4. l162. Please specify the actual activities used in the simulation, even if they correspond to the values in NEMA. The NEMA standard specifies a background activity in the IQ-phantom of 5.3 kBq/ml, a sphere concentration of 21.2 kBq/ml, and 116 MBq in the line source inside the NEMA scatter phantom.
P5. Subsection 2.4 “Sensitivity”: I am confused about how the measurements compare to the NEMA standard. What were the simulation parameters and how are randoms corrected? The standard specifies that the measurements should be made with a 70 cm line source with a known activity amount. Corrections for decay during the acquisition should also be made. What distance do you move the point source axially through the gantry? Are distances the same for the reference and panel systems?
P6-7. Subsections 3.1.1+3.1.2: I find the discussion about the impact of number of iterations, artefacts and reconstructed appearance of the sphere source too long. It is well know that convergence of iterative reconstructions depend on noise, contrast, and number of iterations. On clinical systems, reconstructions are often regularized and various corrections are typically implemented in the system model to help correct for some of the artefacts that are observed in this study with basic MLEM. I believe figure 2 can be left out. However, figure 3 is good. I suggest that you shorten the discussion of MLEM-challenges and move it to the “Discussion”-section. Maybe also discuss it in context of why FBP cannot be used for assessment of spatial resolution.
P6. L223. How can the number of iterations affect the standard NEMA method for spatial resolution measurements? FBP is, as previously mentioned, specified for spatial resolution measurements in the NEMA standard.
P7. Figure 4+ P8 figure 5. I agree that these figures are difficult to read. They are not bad, but maybe it would be more illustrative with a picture of the reconstructed point source in each pixel showing their0 extent in both x- and y-directions.
P9 subsection 3.3 “Sensitivity”: All figures have the expected triangular shape. However, the figure for the reference scanner does not resemble published numbers (cf. Performance Characteristics of the Digital Biograph Vision PET/CT System. Joyce van Sluis, Johan de Jong, Jenny Schaar, Walter Noordzij, Paul van Snick, Rudi Dierckx, Ronald Borra, Antoon Willemsen and Ronald Boellaard. Journal of Nuclear Medicine July 2019, 60 (7) 1031-1036; DOI: https://doi.org/10.2967/jnumed.118.215418). I suggest that numbers are recalculated to match the NEMA specification, thus making comparisons to clinical systems easier.
Furthermore the total system sensitivity is not reported. Please add that value as well, e.g. in table 1 p12 in place of the reported "peak sensitivity".
P10. Figures 7+8. The figure text blocks data, which makes the figures difficult to read. I suggest redesigning the figures.
P11 l363. Discussing the scintillator volume does not make sense to me unless you include the different gantry openings in your discussing.
Author Response
The manuscript discusses the performance of a flat panel limited angle TOF-PET system as a function of crystal size and length, gaps between crystals, crystal read-out strategy and time-of-flight temporal resolution. The discussion is based on GATE simulations and CASToR MLEM reconstructions. Results are compared to simulations of a Siemens Biograph Vision clinical PET/CT system. The proposed flat panel PET/CT system was introduced in a paper published in IEEE Transactions on Radiation and Plasma Medical Sciences by the same research group in 2021. The present manuscript adds data and simulations to the previous work.
Thank you for this in-depth review. Your insight has proven valuable in improving this article. We have carefully considered your comments and suggestions, and revised the manuscript accordingly.
Comment 1: P1: l21-23: It is ok to use the Biograph Vision Scanner as reference for the performance of a modern clinical PET/CT system, but the Vision scanner is not “the state-of-the-art” clinical system. Other systems, e.g. the United Imaging uMI Panorama PET/CT have better performance specifications, including CTR (189 ps) (cf. “Performance Evaluation of the uMI Panorama PET/CT System in Accordance with the National Electrical Manufacturers Association NU 2-2018 Standard. Guiyu Li, Wenhui Ma, Xiang Li, Weidong Yang, Zhiyong Quan, Taoqi Ma, Junling Wang, Yunya Wang, Fei Kang and Jing Wang. Journal of Nuclear Medicine April 2024, 65 (4) 652-658; DOI: https://doi.org/10.2967/jnumed.123.265929). I therefor suggest rephrasing of this sentence.
Response 1: Thank you for bringing our attention to this recent publication. We removed references to the Vision scanner being the state-of-the-art (P1: l:10, 14, 22, P13: l:419).
Comment 2: P2-3: subsection “2.1 Scanner parameters”: I miss some details in this section on how corrections for scatter, attenuation, randoms, and normalisation are implemented. In theory, data are directly available from the simulation. However, image quality depends on PET corrections and implementation details are important.
Response 2: For this study we only considered true events, so no corrections for randoms or scatters were performed. The description in section 2 was now expanded to make it clearer (P2: l:84). Normalisation was performed using built-in functionality of CASToR, except for the case discussed in section 2.3, where additional corrections were performed.
Comment 3: P3: l111-119: Please, specify explicitely both the axial and transaxial FOVs of the reference scanner. Obviously the volume and cost of scintillators in the reference system are higher than on the panel system , but that seems to me to be mainly because of a larger gantry opening – 78 cm vs. 40 cm.
Response 3: The axial FOV and bore radius was explicitly added (P3: l:125). The possibility of reducing the distance between the panels is a specific benefit of flat panel PET, in large part enabling the performance demonstrated. The exact effects on imaging performance by moving the panels further apart (without changing the size of the panels) is part of our future research.
Comment 4: P3: l131: Why is it important to avoid randoms in the simulation, if the contribution to the sinograms are completely known from simulation? Again, I miss more details on the corrections.
Response 4: We hope we sufficiently addressed the corrections in response 2. In this case the reason for wanting to avoid random events was due to computational and storage burden that they pose in the study.
Comment 5: P3: l133: MLEM reconstructions are used for evaluating spatial resolution, which is not according to the NEMA NU-2 2018 specifications, which specify FBP. I guess FBP is not used because of gaps in the projection set between dectector panels. Nonetheless, please give some details on why FBP was not used.
Response 5: That is correct, it is due to the geometry of the scanner. Explanation has been added (P4: l: 145).
Comment 6: P4: l147: Convergence and artefacts from the MLEM algorithm will depend on the number of iterations, signal-to-background contrast, and statistics. Please specify the acquisition time?
It is specified that the background activity was 28.8 kBq in a volume of 4*4*4 cm3. It is also specified that the source activity was 70 Bq in a voxel size of 0.8 mm (should be specified as 0.8x0.8x0.8 mm3). What was the reconstructed background voxel activity concentration (kBq/ml)? The source activity is 70 Bq. What was the reconstructed source activity voxel concentration (kBq/ml)? I cannot from these numbers alone reproduce a source-to-background ratio of 8. Maybe the specified source-to-background ratio is what you measure on your reconstructed images?
I suggest that you rephrase and clarify details.
Response 6: Acquisition time was varied from 30 h to 130 h for different scanner configurations, to compensate for the differences in their sensitivity. This has been added (P5: l:166). Voxel sizes are now specified in 3D (P4: l:147, P5: l:172, P5: l:180, P6: l:218, 230). Background activity concentration was 0.45 kBq/ml and source activity concentration was 16.7 MBq/ml. They are now included in the text (P4: l:143, 162) . You are correct, the ratio is what was measured in reconstructed images (peak height minus background divided by background or in other words from the equation (1) A/C), which is also affected by the spatial resolution of the system, which is much lower (worse) than the point source radius. This is now specified in (P4: l:163).
Comment 7: P4. l162. Please specify the actual activities used in the simulation, even if they correspond to the values in NEMA. The NEMA standard specifies a background activity in the IQ-phantom of 5.3 kBq/ml, a sphere concentration of 21.2 kBq/ml, and 116 MBq in the line source inside the NEMA scatter phantom.
Response 7: Activities are now specified (P5: l:176).
Comment 8: P5. Subsection 2.4 “Sensitivity”: I am confused about how the measurements compare to the NEMA standard. What were the simulation parameters and how are randoms corrected? The standard specifies that the measurements should be made with a 70 cm line source with a known activity amount. Corrections for decay during the acquisition should also be made. What distance do you move the point source axially through the gantry? Are distances the same for the reference and panel systems?
Response 8: In this case there is again no random correction (response 2). The source is moved in 1 mm increments, and the procedure was the same for all scanners. The procedure doesn’t follow the NEMA standard, as the simulation allows for attenuation free measurement, where the exact number of true events is known. There is also no correction for decay since the simulated events are back-to-back gammas and not beta+ decays, as is written in section 2 so the activity doesn’t change during the measurement.
Comment 9: P6-7. Subsections 3.1.1+3.1.2: I find the discussion about the impact of number of iterations, artefacts and reconstructed appearance of the sphere source too long. It is well know that convergence of iterative reconstructions depend on noise, contrast, and number of iterations. On clinical systems, reconstructions are often regularized and various corrections are typically implemented in the system model to help correct for some of the artefacts that are observed in this study with basic MLEM. I believe figure 2 can be left out. However, figure 3 is good. I suggest that you shorten the discussion of MLEM-challenges and move it to the “Discussion”-section. Maybe also discuss it in context of why FBP cannot be used for assessment of spatial resolution.
Response 9: Section 3.1.1. (Effects of background activity) has been removed along with figure 2. Some of the text was rephrased and moved to (P6: l:235, P12: l:373)
Comment 10: P6. L223. How can the number of iterations affect the standard NEMA method for spatial resolution measurements? FBP is, as previously mentioned, specified for spatial resolution measurements in the NEMA standard.
Response 10: Indeed, the number of iterations cannot affect the standard NEMA method using FBP. The statement was removed (see also Response 9).
Comment 11: P7. Figure 4+ P8 figure 5. I agree that these figures are difficult to read. They are not bad, but maybe it would be more illustrative with a picture of the reconstructed point source in each pixel showing their0 extent in both x- and y-directions.
Response 11: We attempted many ways to visualize these results, including 2D and 3D visualizations of point spread as suggested. When comparing different ways to present the data, the presentation as in Figures 4 and 5 (now 3 and 4) was chosen as it combines both the precise numerical data with at least some visual indication (the color bar), while the more graphical visualizations did not communicate the actual resolution well in the format of printed article.
Comment 12: P9 subsection 3.3 “Sensitivity”: All figures have the expected triangular shape. However, the figure for the reference scanner does not resemble published numbers (cf. Performance Characteristics of the Digital Biograph Vision PET/CT System. Joyce van Sluis, Johan de Jong, Jenny Schaar, Walter Noordzij, Paul van Snick, Rudi Dierckx, Ronald Borra, Antoon Willemsen and Ronald Boellaard. Journal of Nuclear Medicine July 2019, 60 (7) 1031-1036; DOI: https://doi.org/10.2967/jnumed.118.215418). I suggest that numbers are recalculated to match the NEMA specification, thus making comparisons to clinical systems easier.
Furthermore the total system sensitivity is not reported. Please add that value as well, e.g. in table 1 p12 in place of the reported "peak sensitivity".
Response 12:
The sensitivity study deviates from the NEMA standard (see also response 8) since in this study the measured number of events is for the whole system for a specific placement of the point source, as opposed to the NEMA standard, where it’s the number detected from the entire source for a single slice. It also makes use of single slice rebinning to assign counts from oblique LORs to slices, while in our study there is no rebinning.
However, total system sensitivity can still be calculated the same way from the NEMA standard. The table has been modified to include total system sensitivity instead of “peak” and now reflects measured values more apparently. When expressed in terms of total system sensitivity, the refence scanner in our simulation achieved the value of 15.4 kcps/MBq, reasonably close to the value of 16.4 kcps/MBq as measured in the referenced article and to the value of 15.1 kcps/MBq measured in https://doi.org/10.1109/NSSMIC.2018.8824710
Comment 13: P10. Figures 7+8. The figure text blocks data, which makes the figures difficult to read. I suggest redesigning the figures.
Response 13: Figures 7 and 8 (now 6 and 7) have both been redesigned to not block data anymore.
Comment 14: P11 l363. Discussing the scintillator volume does not make sense to me unless you include the different gantry openings in your discussing.
Response 14: Panel distance and gantry opening diameter has been added to the discussion (P11: l:359)
Reviewer 2 Report
Comments and Suggestions for Authors
Dear authors
The manuscript submitted is of interest. Below are my comments that in my point of view may improve the submitted manuscript
Comment 1
Introduction. The authors should consider additionally incorporating the following new references and discuss their novelty with respect to them
Meysam Dadgar, Jens Maebe, Maya Abi Akl, Boris Vervenne and Stefaan Vandenberghe "A simulation study of the system characteristics for a long axial FOV PET design based on monolithic BGO flat panels compared with a pixelated LSO cylindrical design" EJNMMI Physics (2023) 10:75, https://doi.org/10.1186/s40658-023-00593-0
Stefaan Vandenberghe, Florence M. Muller, Nadia Withofs, Meysam Dadgar, Jens Maebe, Boris Vervenne, Maya Abi Akl, Song Xue, Kuangyu Shi, Giancarlo Sportelli, Nicola Belcari, Roland Hustinx, Christian Vanhove, Joel S. Karp "Walk‑through flat panel total‑body PET: a patient‑centered design for high throughput imaging at lower cost using DOI‑capable high‑resolution monolithic detectors" European Journal of Nuclear Medicine and Molecular Imaging (2023) 50:3558–3571 https://doi.org/10.1007/s00259-023-06341-x
Comment 2
Materials and Methods
Lines 74-76. The authors did not choose to examined the scintillator optical properties, but a small discussion of the scintillator rise and decay times effect in the image should be delivered. Time values for LYSO and LSO can be found in literature. For instance at Gundacker, R.M. Turtos, E. Auffray, P. Lecoq "Precise rise and decay time measurements of inorganic scintillators by means of X-ray and 511 keV excitation" Nuclear Inst. and Methods in Physics Research, A 891 (2018) 42–52, https://doi.org/10.1016/j.nima.2018.02.074
Or any other reference the authors wish to use.
Comment 3
Materials and Methods
Scanner parameters. The authors should make a small discussion how the sparse (empty space) between the crystals may affect the image SNR, since for flat panel detectors the active detector area plays a role in the final image SNR. I understand that in PET image SNR is holistic affected by several factors.
Comment 4
Materials and methods
Line 130
Do you mean "colineariry" instead of "accolineariry"?
Comment 5
Materials and Methods
Lines 149-151
Please either incorporate the background cube in Figure 1, or make a special figure for it
Comment 6
Materials and methods
Line 201. The authors mentioned Figure 9 too early. Please re-arrange the text of the figures. Alternatively you may delete the sentence.
Comment 7
Results
Please provide some indicative values of A as well as values of variances shown in equation 1. If it is difficult please provide a range.
Comment 8
Results
If feasible please provide the uncertainty of the simulation
Comment 9
Results
Spatial Resolution
It is not clear to me how you measure spatial resolution. You have used spheres of different dimensions and determined which one is visible , or you measure the imaged dimensions of the 0.1 mm radius dots shown in figure 1? Or something else? Please specify.
Comment 10
Results
Line 237.
Please clarify the type of the source (the 0.1 mm radius?)
Comment 11
Discussion
A short sentence should be added commenting why image quality degrades with decreasing crystal length. Please see lines 317-318 of the manuscript.
kind regards
Author Response
Dear authors
The manuscript submitted is of interest. Below are my comments that in my point of view may improve the submitted manuscript
Thank you for this review. It has brought great perspective for the improvement of the article.
Comment 1:
Introduction. The authors should consider additionally incorporating the following new references and discuss their novelty with respect to them
Meysam Dadgar, Jens Maebe, Maya Abi Akl, Boris Vervenne and Stefaan Vandenberghe "A simulation study of the system characteristics for a long axial FOV PET design based on monolithic BGO flat panels compared with a pixelated LSO cylindrical design" EJNMMI Physics (2023) 10:75, https://doi.org/10.1186/s40658-023-00593-0
Stefaan Vandenberghe, Florence M. Muller, Nadia Withofs, Meysam Dadgar, Jens Maebe, Boris Vervenne, Maya Abi Akl, Song Xue, Kuangyu Shi, Giancarlo Sportelli, Nicola Belcari, Roland Hustinx, Christian Vanhove, Joel S. Karp "Walk‑through flat panel total‑body PET: a patient‑centered design for high throughput imaging at lower cost using DOI‑capable high‑resolution monolithic detectors" European Journal of Nuclear Medicine and Molecular Imaging (2023) 50:3558–3571 https://doi.org/10.1007/s00259-023-06341-x
Response 1:
The studies were found to be of interest regarding the development of flat panel PET detectors, which they approach from a different perspective and were included as references. (Page 2 line 41-44, page 14 line 471-476.
The novelty of manuscript under review is that it focuses on the improvements in CTR and reduction in cost by using smaller total crystal volume, compared to flat panels using BGO and DOI in the two references.
Comment 2:
Materials and Methods
Lines 74-76. The authors did not choose to examined the scintillator optical properties, but a small discussion of the scintillator rise and decay times effect in the image should be delivered. Time values for LYSO and LSO can be found in literature. For instance at Gundacker, R.M. Turtos, E. Auffray, P. Lecoq "Precise rise and decay time measurements of inorganic scintillators by means of X-ray and 511 keV excitation" Nuclear Inst. and Methods in Physics Research, A 891 (2018) 42–52, https://doi.org/10.1016/j.nima.2018.02.074
Or any other reference the authors wish to use.
Response 2:
The discussion on L(Y)SO crystal properties was now expanded on page 2 lines 48-50.
Comment 3:
Materials and Methods
Scanner parameters. The authors should make a small discussion how the sparse (empty space) between the crystals may affect the image SNR, since for flat panel detectors the active detector area plays a role in the final image SNR. I understand that in PET image SNR is holistic affected by several factors.
Response 3:
The discussion on sparse design has been added on page 3, lines 96-102 along with a reference on page 14 line 494. Sparse configurations lead to reduced geometric coverage and sensitivity. Studies have been performed using conventional cylindrical scanners, which show that image quality is preserved despite the sparse configuration.
Comment 4:
Materials and methods
Line 130
Do you mean "colineariry" instead of "accolineariry"?
Response 4:
The text was modified to be more clear: non-colinearity (GATE parameter accolinearity) of the source was set to 0.5°. (page 4, line 141)
Comment 5:
Materials and Methods
Lines 149-151
Please either incorporate the background cube in Figure 1, or make a special figure for it
Response 5:
Figure 1 has been modified to include a background cube.
Comment 6:
Materials and methods
Line 201. The authors mentioned Figure 9 too early. Please re-arrange the text of the figures. Alternatively you may delete the sentence.
Response 6:
The sentence was removed.
Comment 7:
Results
Please provide some indicative values of A as well as values of variances shown in equation 1. If it is difficult please provide a range.
Response 7:
Values of A are arbitrary and correspond to the height of the peak of the reconstructed source, the range of values for the CS3_L10_CTR75 for example was 0.006 to 0.02. Variances were recalculated into FWHM and are shown as results for spatial resolution in figures 2, 3 and 4.
Comment 8:
Results
If feasible please provide the uncertainty of the simulation
Response 8:
The uncertainty of the simulation was in part shown in the image quality section (3.2), where the simulations were performed 10 times. For the spatial resolution the uncertainty was estimated from the covariance of the fit, with the deviation of the resolution parameters ranging from 0.002 to 0.03, which would make the error bars mostly not visible for example on bottom of figure 3. Uncertainty was not included for the sake of readability and not increasing the complexity of the figures.
Comment 9:
Results
Spatial Resolution
It is not clear to me how you measure spatial resolution. You have used spheres of different dimensions and determined which one is visible , or you measure the imaged dimensions of the 0.1 mm radius dots shown in figure 1? Or something else? Please specify.
Response 9:
We have measured the imaged dimensions of the 0.1 mm radius dots shown in figure 1: the spatial resolution measured by fitting the function from equation 1 to the reconstructed source and taking the three variances as resolution in each dimension.(Page 4, line 148 in revised manuscript).
Comment 10:
Results
Line 237.
Please clarify the type of the source (the 0.1 mm radius?)
Response 10:
The sources used for spatial resolution study were spheres of 0.1 mm radius (page 3 line 137 and page 6 line 227 in revised manuscript).
Comment 11:
Discussion
A short sentence should be added commenting why image quality degrades with decreasing crystal length. Please see lines 317-318 of the manuscript.
Response 11:
The comment on degradation has been added on page 9 line 310.
Round 2
Reviewer 2 Report
Comments and Suggestions for Authors
Thank you for addressing my comments.